# Competitiveness of Agricultural Products in the Eurasian Economic Union

**Vlada Maslova** [1,*]**, Natalya Zaruk** [2]**, Clemens Fuchs** [3] **and Mikhail Avdeev** [1]

[1]  Federal State Budgetary Scientific Institution "Federal Research Center of Agrarian Economy and Social Development of Rural Areas—All-Russian Research Institute of Agricultural Economics", 123007 Moscow, Russian Federation; avdeevmihail@mail.ru

[2]  Federal State Budgetary Educational Institution of Higher Education "Russian State Agrarian University-Moscow Timiryazev Agricultural Academy", Institute economics and management in agribusiness, 127550 Moscow, Russian Federation; zaruk84@bk.ru

[3]  Faculty Agriculture and Food Sciences, University of Applied Sciences Neubrandenburg, 17033 Neubrandenburg, Germany; cfuchs@hs-nb.de

*  Correspondence: maslova_vlada@mail.ru; Tel.: +7-499-195-60-28

**Abstract:** This article discusses the outcomes of a quantitative analysis using econometric panel data models of the competitiveness of grains in the countries of the Eurasian Economic Union (EAEU). The analysis was based on the public authorities' statistics of EAEU countries as well as the United Nations Comtrade Database, which is a repository of official international trade statistics. The results of the analysis allowed us to assess the level of competitiveness of the agricultural products produced in EAEU countries and to determine the extent to which various factors affect the competitiveness. The research conclusions can be used to develop and adjust the agricultural policy in the EAEU.

**Keywords:** agriculture; grain; competitiveness; prices; production volumes; export; import; factor analysis

## 1. Introduction

The Eurasian Economic Union (EAEU) was established in 2015 and now consists of five member states: the Republic of Armenia, the Republic of Belarus, the Republic of Kazakhstan, the Kyrgyz Republic, and the Russian Federation. One of the primary goals of the EAEU is the development of coordinated agricultural policy-making between the member states including measures to ensure the competitiveness of produced agricultural products [1].

One of the main challenges facing the economies of EAEU countries is increasing the competitiveness of products. This issue has become particularly relevant in the context of increasing globalization and cross-country competition. We focused on the identification and assessment of the main factors that influence the competitiveness of the agricultural products in the EAEU countries using a factor analysis of the competitiveness of agricultural products. As a research tool, we used the methodology for determining the integral index of competitiveness for various types of agricultural products for the EAEU countries, as well as correlation regression analysis based on the econometric software package EViews 9.5 (IHS Global Inc., Irvine, CA, USA)

The most important segment of the EAEU agricultural market is the grain market. The EAEU share in the world production of grain was 6.3% in 2017. The EAEU countries produce 13.5% and 17.5% of the world production of wheat and barley, respectively in 2017. The share of EAEU countries in the world trade of wheat and barley was about 20% and 14.3%, respectively in 2017. Therefore, the grain

market was chosen as the object of the factor analysis of competitiveness as a competitive market [2]. In this research, we extensively studied the types of agricultural products such as wheat, barley, and corn.

## 2. Methodology

Many studies by both foreign and Russian scientists have examined the issues of competition and competitiveness. However, a common concept for these definitions as well as a common methodology for calculating the indicators of competitiveness have yet to be developed because competitiveness is a complex concept that includes many significantly different objects and factors. As noted by Fatkhutdinov, competitiveness integrates technological, economic, managerial, marketing, psychological, and other characteristics of the subject and object, which depend on specific market conditions [3]. Another important aspect of the assessment of competitiveness is the level on which the assessment is conducted (micro, meso, or macro). For example, competitiveness can be assessed at the level of the production of goods at a particular enterprise in comparison with other enterprises in the industry, or at the level of the region or country in comparison with the production of similar goods in other regions and countries.

Porter defined the competitiveness of a product, service, or a subject as the ability to act on the market on par with similar goods, services, or competing subjects within market relations [4]. The main criteria for assessing competitiveness is the ability to compete in price and quality, while ensuring a steady increase in production [5]. In this regard, various authors have used the concept of the competitive advantage of the characteristics and properties of a product or brand that result in a certain superiority over their direct competitors [6].

Various methods are available that use various aspects of competitiveness for assessing the competitiveness of goods. One of the most common methods is the assessment of competitiveness based on the calculation of indices of comparative competitive advantages. Among them, the most widely used is the revealed comparative advantage index introduced by Balassa. The index is determined by the ratio of the export of goods to the total export of a country to the world export of this product to the world export of all goods [7,8]. Various authors developed modifications of this index: the Proudman–Redding index, which corrects the distortions that arise when comparing large and small countries by normalizing the Balassa index to the average value for goods for a particular country [9]; the Bowen index, which uses the calculation data on production in the country [10]; and the Lafey index, which considers a country's e exports and imports [11].

A similar analysis using the calculation of the coefficient of comparative competitive advantages was used by Gibba, who determined the competitive advantages of countries involved in world trade of vegetables [12]; Smutka et al. analyzed the effect of product embargo on trade relations between Russia and the Euorpan Union (EU) [13]; and Esquivias analyzed the dynamics of the comparative advantages of the Association of Southeast Asian Nations ASEAN countries and identified their export specialization on the basis of trade flow information [14].

Among other approaches to the definition of competitiveness, one methodology can be singled out based on the evaluation of goods from the point of view of their ability to satisfy consumer demands (the Rosenberg model). This method considers the importance of product characteristics from the consumer viewpoint. This model was used in the analysis by Bronnikova et al. [15] and was used the calculation of the integral indicator, for example, if the assessment of the competitiveness of an object is conducted on the basis of a large group of indicators [16].

These methods can be used to assess the competitiveness of agricultural products. However, in our opinion, for the analysis of product competitiveness at the macrolevel, the most expedient approach is the use of the integral index because a complex indicator synthesizes various product characteristics. This indicator was used to reveal the intercountry competitiveness of the agricultural products of the EAEU countries.

To evaluate the competitiveness of the EAEU countries for various types of agricultural products, we propose a methodology for assessing the competitiveness of products based on the calculation of an integrated indicator that includes five factors: average producer price ($AVP_{ij}$), export price ($EX\_P_{ij}$), output ($Q_{ij}$), share export in production volume ($EX\_Q_{ij}$), price competitiveness factor ($Com\_C_{ij}$) defined as the ratio of average prices of agricultural producers, and import prices ($IM\_P$), considering imports, duties, customs duties, value-added tax (*VAT*), and excise duties [16]. This indicator allows for the assessment of the competitiveness of agricultural products at various levels, from production and realization to the level of mutual (within the EAEU) and foreign trade.

To ensure the comparability of the calculations, each *i* kind of product for each *j* EAEU country must be normalized using the following maximising equation:

$$AVP_{ij} = \frac{AVP_{ij}}{max\left\{AVP_{ij}\right\}} \tag{1}$$

where $AVP_{ij}$ is the average producer price,

$$EX\_P_{ij} = \frac{EX\_P_{ij}}{max\left\{EX\_P_{ij}\right\}} \tag{2}$$

where $EX\_P_{ij}$ is the export price,

$$Q_{ij} = \frac{Q_{ij}}{max\left\{Q_{ij}\right\}} \tag{3}$$

$Q_{ij}$ is the output,

$$EX\_Q_{ij} = \frac{EX\_Q_{ij}}{max\left\{EX\_Q_{ij}\right\}} \tag{4}$$

$EX\_Q_{ij}$ is the share export in production volume, and

$$Com\_C_{ij} = \frac{Com\_C_{ij}}{max\left\{Com\_C_{ij}\right\}} \tag{5}$$

$Com\_C_{ij}$ is the price competitiveness factor.

To calculate the integral indicator of competitiveness, these indicators were divided into two groups. The first group of indicators determines the competitiveness of products based on price factors ($AVP_{ij}$, $EX\_P_{ij}$, and $Com\_C_{ij}$); therefore, products become more competitive with a decrease in these indicators. The second group of indicators characterizes the competitiveness of products based on the production volumes and the share of exports ($Q_{ij}$ and $EX\_Q_{ij}$); therefore, the products become more competitive with an increase in these indicators.

The calculations were based on these multidirectional factors. To ensure their comparability, an additive model of deterministic factor analysis was used. On the basis of normalized factors, an integral indicator of competitiveness was determined for each type of agricultural product of the EAEU states ($INT\_C_{ij}$):

$$INT\_C_{ij} = \left\{1 - \left(AVP_{ij} + EX\_P_{ij} + \left(1 - Q_{ij}\right) + \left(1 - EX\_Q_{ij}\right) + Com\_C_{ij}\right)/5\right\} \tag{6}$$

The comparison of the competitiveness of EAEU countries was based on the values of this indicator for certain products for a particular year. The higher the integrated indicator of competitiveness, the more competitive the products in both the domestic and foreign markets.

On the basis of the integral index of competitiveness, two tasks are solved. The first task was to compare and rank the EAEU countries in terms of product competitiveness. The second task was to assess the degree of influence of the indicated factors ($AVP_{ij}$, $EX\_P_{ij}$, $Com\_C_{ij}$, $Q_{ij}$, and $EX\_Q_{ij}$)

on the level of competitiveness of various types of products in each EAEU country as well as ranking these factors.

　　The first task was to compare and rank the EAEU countries in terms of product competitiveness. To solve this problem, we constructed a regression multi-country model of the first type:

$$INT_{\_w} = b_0 + b_1 RF_{ij} + b_2 RB_{ij} + b_3 KZ_{ij} + b_4 AM_{ij} + b_5 KG_{ij} + \varepsilon \tag{7}$$

where $b_0$ is the conditional start of the regression model (constant); $b_i$ is the coefficient of pure regression; $INT_{\_w}$ is the the weighted average integral indicator of competitiveness; $RF_{ij}$, $RB_{ij}$, $KZ_{ij}$, $AM_{ij}$, and $KG_{ij}$ are integral indicators of competitiveness of *i* products in *j* state (Russia, Belarus, Kazakhstan, Armenia, and Kyrgyzstan), respectively; and $\varepsilon$ is the random error of the regression model. This model enables the assessment of the competitiveness of the EAEU countries over a long period of time.

　　One dependent variable in this model wisas the weighted average indicator of the competitiveness of grain and its processing products, in general, for the EAEU $INT_{\_w}$. To determine this indicator, we had to calculate the integral indicators of competitiveness ($RF_{ij}$, $RB_{ij}$, $KZ_{ij}$, $AM_{ij}$, and $KG_{ij}$) for *i* type of product *j* of the country using Equation (6). Then, we calculated the weighted average indicator, $INT_{\_w}$, as weights used for share *j* of the state in the total volume of gross grain production in the EAEU.

　　The second task was to assess the degree of influence of the indicated factors ($AVP_{ij}$, $EX\_P_{ij}$, $Com\_C_{ij}$, $Q_{ij}$, and $EX\_Q_{ij}$) on the level of competitiveness of various types of products in each EAEU country as well as ranking these factors. The method includes: (1) calculating the integral indicator of product competitiveness in the EAEU countries, (2) cross-country regression analysis of the integral indicator of dynamic competitiveness, and (3) correlation and regression analysis to identify the degree of influence of $AVP_{ij}$, $EX\_P_{ij}$, $Com\_C_{ij}$, $Q_{ij}$, and $EX\_Q_{ij}$ on the integral indicator of product competitiveness in the EAEU countries.

　　The degree of influence of price determinants, factors of the volume of production, and export of grain on the integral indicator of its competitiveness were estimated on the basis of the coefficients of pure regression ($b_n$) for the *j* state and *i* type of output of the grain market using the models of multiple regression:

$$INT\_C = b_0 + b_1 AVP_{ij} + b_2 EX\_P_{ij} + b_3 Com\_C_{ij} + b_4 Q_{ij} + b_5 EX\_Q_{ij} + \varepsilon \tag{8}$$

where *INT_C* is the integral indicator of competitiveness; *AVP* is the average producer prices of grain products in the EAEU countries, US dollars; *EX_P* is the export prices for grains and products of its processing in the EAEU countries, US dollars; *Com_C* is the coefficient of price competitiveness; *Q* is the volume of cereal production from the member states of the EAEU countries, in thousand tons; and *EX_Q* is the share of grain exports from the EAEU countries in total production.

　　Thus, our method allows, first, identifying which of the EAEU countries is the most competitive in various agricultural markets, and, second, determining which of the factors has the greatest influence on the formation of competitive advantages in each EAEU country.

　　During the research, a set of methods was used: a monographic study of the research subject, expert assessments, statistical methods, and econometric modeling. Through the econometric model, the main quantitative relationships between the analyzed economic phenomena and processes can be described [17]. The construction of competitiveness models and the evaluation of their parameters were conducted in a package in the applied econometric program Eviews 9.5 (IHS Global Inc., Irvine, CA, USA).

　　We used the annual data of the state statistical offices of the EAEU countries and the international trade database United Nations (UN) Comtrade on the volume of production, export-import operations, and prices for wheat, barley, corn, wheat flour, and pasta by the EAEU countries for the period of 2010–2016 in this study.

## 3. Results and Discussion

### 3.1. Analysis of Factors of Competitiveness

The first factor investigated was the producer prices for grain [18]. In 2016, the lowest producer price for each of the products considered was in the Republic of Kazakhstan: wheat was USD $105/ton, barley was USD $77/ton, and corn for grain was USD $114/ton. The highest prices for wheat and barley were registered in the Republic of Armenia at USD $231/ton, USD $212/ton, respectively.

The second factor was export prices. In addition to the cost of the goods, the export price includes transportation, insurance, and indirect taxes. In 2016, the most competitive export prices in the domestic market of the EAEU for wheat were in the Republic of Belarus at USD $131/ton, barley was USD $95/ton in the Republic of Kazakhstan, and corn in the Russian Federation was USD $929/ton.

The third factor was the coefficient of price competitiveness [2]. The analysis of the coefficient of price competitiveness eanbled determining that the Republic of Belarus, the Republic of Kazakhstan, and the Russian Federation have the greatest competitive advantages among the EAEU states. In addition to price factors, an important indicator is the volume of production and exports. These factors may be minimal at a competitive price, but they do not increase the competitiveness of products.

The next investigated factor was the volume of grain production. In 2016, the gross grain harvest in the EAEU reached a record value of 151 million tons. This figure was mainly due to the increase in grain production in the Russian Federation and the Republic of Kazakhstan, which occurred due to the increase in grain yields and the growth in acreage [19].

The gross wheat harvest in the EAEU has fluctuated significantly from 51 million tons in 2012, which was due to a catastrophic drought in Russia, to almost 92 million tons in 2016. The growth from 2012 to 2016 amounted to almost 80% (Figure 1). As a result, the EAEU ranked third place in the world for wheat production (after the EU at 145 million tons and China at 129 million tons). The maximum growth in production was recorded in Russia where it was 37% more in 2016 than the average for the previous five years (2011–2015). In Armenia, the increase was almost 17% and in Kazakhstan, it was 2%. In Belarus and Kyrgyzstan, the gross wheat harvest was slightly lower than the average annual figures for 2011–2015.

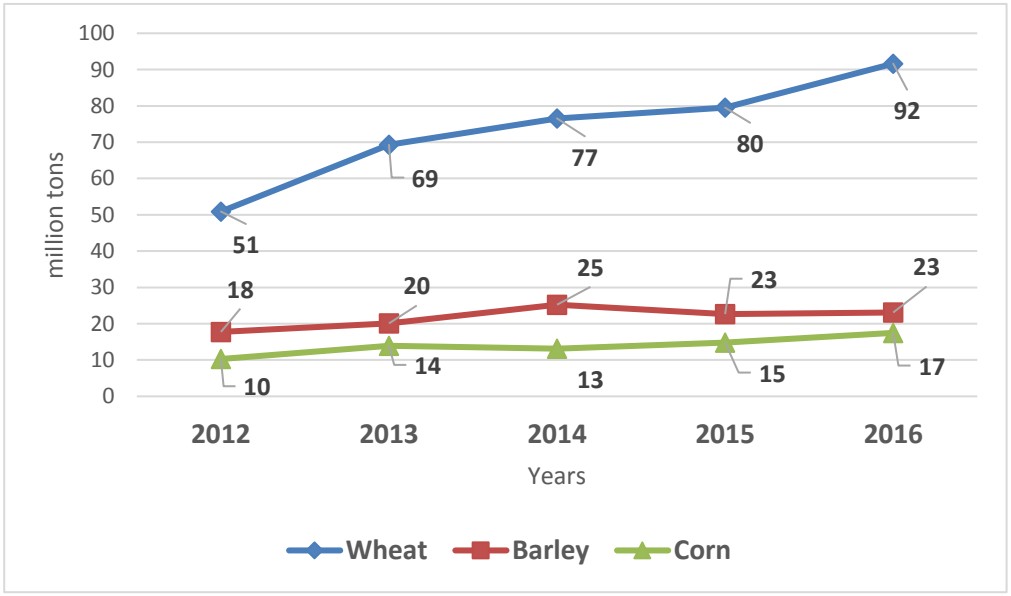

**Figure 1.** Gross collections of wheat, barley, and corn for the grain and volume production in the EAEU countries from 2012 to 2016, in million tons. Compiled from the statistical services of the EAEU countries [20–24].

Gross collections of barley from 2012 to 2016 increased by 30% as a whole in the EAEU, which amounted to 23 million tons in 2016. The largest increases in production were registered in Armenia, Kazakhstan, and Kyrgyzstan. In contrast, in Belarus, the gross barley harvests in the period under review decreased by 25%. High growth rates of gross revenues were observed for corn for grain (an increase of 70%), where the volume of its production in the EAEU in 2016 reached 17.5 million tons. The main producers of grain in the EAEU are Russia, Belarus, and Kazakhstan, and their share in total grain production is almost 99%. The leader in terms of the volume of grain production and its processing products among the states of the EAEU is the Russian Federation.

The fifth factor affecting the competitiveness of products is the volume of exports, or more precisely, the share of exports in production volume, which is the export concentration index. When comparing the share of exports in production, in 2016, the Russian Federation had the highest share of exports in the production volume for wheat (35%), the Republic of Kazakhstan for barley (24%), and the Russian Federation maize for grain (35%) (Figure 2).

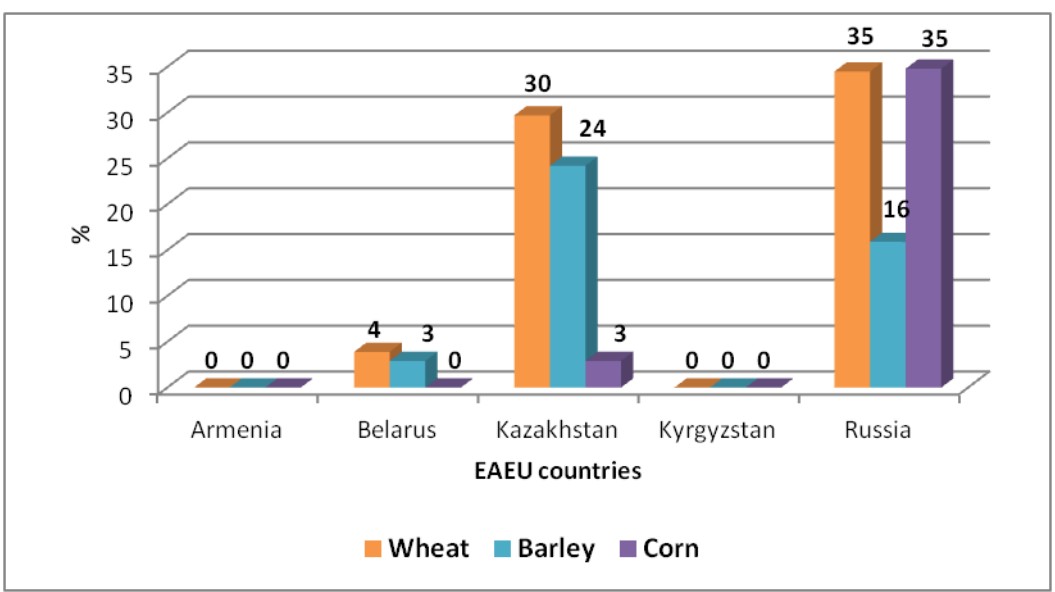

**Figure 2.** Share of exports in the volume of the production of wheat, barley, and corn in the EAEU countries, 2016, %. Source: compiled from UNComtrade and EEC data [20–25].

*3.2. Calculation of the Integral Indicator*

Integral indicators of grain competitiveness on the domestic EAEU market are determined on the basis of average producer prices, export prices, production volumes, and share of export of products. the price competitiveness coefficients are presented in Table 1.

**Table 1.** Integral indicator of the competitiveness of grain in the EAEU countries in 2016, %.

| Product | Armenia | Belarus | Kazakhstan | Kyrgyzstan | Russian Federation |
|---------|---------|---------|------------|------------|--------------------|
| Wheat | 0 | 62.0 | 73.0 | 0 | 100 |
| Barley | 0 | 55.2 | 88.3 | 0 | 100 |
| Corn | 0 | 4.2 | 17.1 | 0 | 100 |

For wheat, barley, and corn, the Russian Federation had the highest integral index. The Republic of Armenia and the Kyrgyz Republic were not competitive in the grain market. Aspreviously mentioned, almost all grain production in the EAEU is concentrated in three countries: Russia, Kazakhstan, and Belarus. Therefore, the object of the correlation-regression analysis of the competitiveness factors was the grain products of these three countries. Next, we conducted a cross-country regression analysis

of the dynamics of the integral indicator of competitiveness (2010–2016) and a correlation-regression analysis of the degree of influence of $AVP_{ij}$, $EX\_P_{ij}$, $Com\_C_{ij}$, and $Q_{ij}$, $EX\_Q_{ij}$ on the integral indicator in the EAEU countries for wheat, barley, and corn in Russia, Kazakhstan, and Belarus.

## 4. Regression Analysis of Grain Competitiveness Factors

### 4.1. Wheat

Reflecting the dependence of the integrated indicator of wheat competitiveness in the EAEU on the parameters of the competitive advantages of individual EAEU countries, the multiple linear regression model takes the form

$$
\begin{array}{cccccccc}
INT\_w = & 0.062 & + & 0.65RF & - & 0.033RB & + & 0.21KZ \\
& (0.019) & & (0.038) & & (0.019) & & (0.019)
\end{array}
\tag{9}
$$

The values of standard errors are indicated in parentheses at a 5% level of statistical significance.

The parameters of the obtained econometric model show that the most competitive in the 2010–2016 period was wheat produced in the Russian Federation (the largest coefficient of pure regression with variable RF), followed by the Republic of Kazakhstan and the Republic of Belarus (Table 2).

**Table 2.** The results of the comparison of EAEU countries in terms of wheat competitiveness based on regression analysis from 2010 to 2016.

| Factor | Coefficient | Standard Error | *t*-Statistic | Prob (F-Statistic) |
|---|---|---|---|---|
| | | Wheat | | |
| Constant | 0.062 | 0.0014 | 14.811 | 0.022 |
| Russian Federation (RF) | 0.65 | 0.0018 | 6.931 | 0.041 |
| Republic of Belarus (RB) | –0.033 | 0.0017 | –1.232 | 0.028 |
| Republic of Kazahstan (KZ) | 0.21 | 0.0010 | 4.143 | 0.034 |
| $R^2$ = 78%, F = 122.12, Prob (F) = 0.001. | | | | |

Using correlation-regression analysis, we estimated the strength and direction of the influence of price and non-price factors on the competitiveness of wheat in each individual country of the EAEU. The results of the analysis are presented in Table 3.

The data in Table 3 show that the level of wheat competitiveness in the Russian Federation was most influenced by factors such as production volume and average producer prices. The gross harvest of wheat in farms of all categories during 2010–2016 increased by almost 80% from 41.5 million tons to 73.3 million tons. Along with the growth in production, volumes of wheat supply to foreign markets also increased; in relative terms, this figure more than doubled. However, the outstripping growth in wheat production (supply), when compared with the dynamics of demand in domestic and foreign markets, exerts pressure on both the domestic and export prices of agricultural producers. In particular, average domestic wheat prices reached peak values in 2013 at USD \$244/ton, before the price steadily decreased to USD \$132/ton in 2016. The same trend was observed in the export prices for Russian wheat. This strengthens Russia's competitive position in foreign markets.. The price level for Russian wheat is much lower than in the leading export countries of this product in the world. For example, the average export price for wheat produced in Russia in 2016 was USD \$166.5/ton, and in the USA, it was USD \$224.9/ton, and USD \$228.5/ton in Canada [25].

**Table 3.** Results of the regression analysis of competition factors of wheat in Russia, Kazakhstan, and Belarus from 2010 to 2016.

| Factor | Coefficient | Standard Error | *t*-Statistic | Prob (F-Statistic) |
|---|---|---|---|---|
| Russian Federation | | | | |
| Constant | −0.51 | 0.084 | −1.80 | 0.032 |
| Average prices (APV) | −0.81 | 0.021 | 0.92 | 0.041 |
| Quantity (Q) | 1.14 | 0.023 | −0.48 | 0.006 |
| Export prices (EX_P) | −0.41 | 0.018 | 1.92 | 0.011 |
| Share of export (EX_Q) | −0.39 | 0.027 | −0.24 | 0.008 |
| Coefficient of competition (Com_C) | −0.54 | 0.0103 | −1.154 | 0.030 |
| $R^2$ = 98%, F = 32.32, Prob (F) = 0.016. | | | | |
| Republic of Belarus | | | | |
| Constant | 0.26 | 0.026 | 0.08 | 0.033 |
| Average prices (APV) | −0.16 | 0.014 | −1.18 | 0.029 |
| Quantity (Q) | 0.09 | 0.019 | −0.50 | 0.064 |
| Export prices (EX_P) | −0.07 | 0.007 | −1.12 | 0.031 |
| Share of export (EX_Q) | 0.14 | 0.005 | 2.87 | 0.041 |
| Coefficient of competition (Com_C) | −0.17 | 0.007 | −2.60 | 0.049 |
| $R^2$ = 90.5%, F =21.74, Prob F = 0.017. | | | | |
| Republic of Kazahstan | | | | |
| Constant | −0.66 | 0.044 | −1.50 | 0.037 |
| Average prices (APV) | 0.48 | 0.022 | 2.18 | 0.008 |
| Quantity (Q) | 0.83 | 0.027 | 0.16 | 0.009 |
| Export prices (EX_P) | −0.32 | 0.16 | 5.16 | 0.00 |
| Share of export (EX_Q) | 0.08 | 0.019 | 0.48 | 0.0065 |
| Coefficient of competition (Com_C) | −0.09 | 0.022 | −0.37 | 0.0073 |
| $R^2$ = 84.7%, F (s) = 5.55, Prob F = 0.1. | | | | |

In the Republic of Belarus, the factors exerting the greatest influence on the competitiveness of wheat included domestic producer prices and the share of exports in the total volume of gross production. In particular, the wheat prices of Belarusian agricultural producers reached the highest values in 2013 at USD $233/ton, which fell steadily in subsequent years, reaching USD $127/ton in 2016. In this case, the gross collections of wheat increased. Thus, in 2016 compared to 2010, the increase was 34.5%. The volumes of export deliveries grew almost 19 times from 5 to 91.6 thousand tons, and the share of exports increased from 0.28% to 3.9%.

In Kazakhstan, the most important factors affecting wheat competitiveness were the average producer prices, export prices, and production volumes. From 2013, the negative dynamics of the average producer prices remained at a level that was almost halved by 2016; the export prices showed a decrease of about 36%. According to the price criteria, Kazakhstan had the greatest competitive advantages among all EAEUcountries. In the period of 2010–2016, Kazakh agricultural producers increased the volume of wheat production from 9.6 to 15 million tons. However, against the background of growth in production volumes, the share of exports decreased. In particular, in 2010, the volume of wheat deliveries from Kazakhstan to foreign markets amounted to 5 million tons, which was more than half of the total production of this crop. This increased in 2012 when exports amounted to 7.5 million tons, about 76% of gross harvest, and then by 2016, this indicator decreased to 4.4 million tons, about 30% of the total volume of production.

*4.2. Barley*

The multiple linear regression model, reflecting the dependence of the integral indicator of barley competitiveness in the EAEU on the parameters of the competitive advantages of the countries studied, takes the form

$$INT\_w = \underset{(0.007)}{-0.024} + \underset{(0.08)}{0.81RF} - \underset{(0.0013)}{0.07RB} + \underset{(0.063)}{0.15KZ} \tag{10}$$

The values of standard errors are indicated in parentheses at a 5% level of statistical significance.

The parameters of the obtained econometric model showed that the barley produced in the Russian Federation had the most competitive advantage in the period from 2010 to 2016 (the largest coefficient of pure regression with variable RF), followed by the Republic of Kazakhstan, then the Republic of Belarus (Table 4).

**Table 4.** The results of the comparison of the level of barley competitiveness of the EAEU countries based on regression analysis from 2010 to 2016.

| Factor | Coefficient | Standard Error | *t*-Statistic | Prob (F-Statistic) |
|--------|------------|----------------|---------------|--------------------|
| | | Barley | | |
| Constant | −0.24 | 0.009 | 12.322 | 0.002 |
| Russian Federation (RF) | 0.81 | 0.048 | 4.435 | 0.017 |
| Republic of Belarus (RB) | 0.07 | 0.047 | 2.222 | 0.016 |
| Republic of Kazahstan (KZ) | 0.15 | 0.043 | 4.983 | 0.015 |
| $R^2$ = 91%, F = 62.12, Prob (F) = 0.01 | | | | |

Using correlation and regression analysis, we estimated the degree of influence of the studied factors on the competitiveness of barley. The results of the analysis are presented in Table 5.

**Table 5.** Results of regression analysis of competition factors of barley in Russia, Kazakhstan, and Belarus from 2010 to 2016.

| Factor | Coefficient | Standard Error | *t*-Statistic | Prob (F-Statistic) |
|--------|------------|----------------|---------------|--------------------|
| | | Russian Federation | | |
| Constant | 0.16 | 0.043 | 0.38 | 0.008 |
| Average prices (APV) | −0.28 | 0.01 | 3.46 | 0.02 |
| Quantity (Q) | 0.32 | 0.02 | 1.43 | 0.021 |
| Export prices (EX_P) | −0.11 | 0.007 | 2.00 | 0.01 |
| Share of export (EX_Q) | 0.07 | 0.009 | −0.80 | 0.046 |
| Coefficient of competition (Com_C) | −0.21 | 0.008 | −2.75 | 0.04 |
| $R^2$ = 91%, F = 20.99, Prob (F) = 0.016. | | | | |
| | | Republic of Belarus | | |
| Constant | 0.13 | 0.032 | 2,01 | 0,001 |
| Average prices (APV) | −0.26 | 0.011 | −1.98 | 0.011 |
| Quantity (Q) | 0.07 | 0.014 | −1.84 | 0.013 |
| Export prices (EX_P) | −0.09 | 0.006 | −1.57 | 0.018 |
| Share of export (EX_Q) | 0.077 | 0.006 | 1.21 | 0.028 |
| Coefficient of competition (Com_C) | −0.16 | 0.006 | −2.57 | 0.005 |
| $R^2$ = 95.5%, F=14,9, Prob F = 0.013. | | | | |

**Table 5.** *Cont.*

| Factor | Coefficient | Standard Error | *t*-Statistic | Prob (F-Statistic) |
|---|---|---|---|---|
| Republic of Kazakhstan | | | | |
| Constant | 0.21 | 0,116 | 2,87 | 0,0005 |
| Average prices (APV) | –0.22 | 0.019 | –0.62 | 0.056 |
| Quantity (Q) | 0.11 | 0.016 | 0.35 | 0.074 |
| Export prices (EX_P) | –0.11 | 0.014 | –0.07 | 0.095 |
| Share of export (EX_Q) | 0.095 | 0.014 | 1.20 | 0.028 |
| Coefficient of competition (Com_C) | –0.009 | 0.012 | –0.02 | 0.099 |
| $R^2$ = 84.7%, F (s) = 5.55, Prob F = 0.01. | | | | |

On the basis of the data presented in Table 5, we concluded that, in the Russian Federation, the most significant factors of barley competitiveness are volume of production, average producer prices, and the coefficient of price competitiveness. In particular, during the period 2010–2016, the volume of barley production in Russia more than doubled from 8.35 to 18 million tons, which is about 80% of the production of this crop as a whole for all EAEU countries. The dynamics of barley producer prices was characterized by a negative trend, as was the case for wheat. The average prices for Russian barley reached the lowest values in 2013 at USD $216/ton, and by the end of 2016, prices had fallen almost twofold to USD $116/ton. Notably, the prices for barley from Russian agricultural producers were the highest among all EAEU countries. The level of export prices in 2016 allowed Russia to confidently compete with the leading countries, France, Australia, and Germany, in terms of barley exports worldwide. Export prices for barley in these countries in 2016 were USD $182/ton, USD $195/ton, and USD $173/ton, respectively, which was much higher than the export prices for Russian barley, the price of which in 2016 was USD $148/ton.

In the Republic of Belarus, the average producer prices and the price competitiveness factor had the greatest impact on the competitiveness of barley. The least significant factor was the volume of production for the period of 2010–2016. The gross harvest of barley in farms of all categories declined by almost 38% from 1.97 million tons to 1.23 million tons in this period.

In Kazakhstan, the competitiveness of barley was mainly influenced by the price factors. The current level of domestic and export prices for barley in Kazakhstan was the lowest in the grain market of the Unified Energy System (USD $77/ton and USD $95/ton, respectively). The coefficient of price competitiveness did not significantly affect the level of competitive advantages of barley in the internal EAEU market.

### 4.3. Corn

The multiple linear regression model, reflecting the dependence of the integrated indicator of corn's competitiveness in the EAEU on the parameters of the competitive advantages of individual countries of the Union, takes the form

$$INT\_w = \underset{(0.0065)}{-0.033} + \underset{(0.035)}{0.92RF} - \underset{(0.05)}{0.05RB} + \underset{(0.009)}{0.15KZ_{ij}} \tag{11}$$

The values of standard errors are indicated in parentheses at a 5% level of statistical significance.

Parameters of the obtained econometric model showed that, from 2010 to 2016, the most competitive country in corn production was in the Russian Federation (the highest coefficient of pure regression with variable RF), followed by the Republic of Kazakhstan and the Republic of Belarus (Table 6).

**Table 6.** The results of the comparison of the EAEU countries in the level of competitiveness of corn based on regression analysis from 2010 to 2016.

| Factor | Coefficient | Standard Error | *t*-Statistic | Prob (F-Statistic) |
|---|---|---|---|---|
| | | Corn | | |
| Constant | −0.033 | 0.014 | 11.313 | 0.000 |
| Russian Federation (RF) | 0.92 | 0.024 | 6.546 | 0.021 |
| Republic of Belarus (RB) | 0.05 | 0.054 | 4.243 | 0.037 |
| Republic of Kazahstan (KZ) | 0.15 | 0.021 | 3.721 | 0.024 |
| $R^2 = 87\%$, F = 13.32, Prob (F) = 0.041. | | | | |

Using correlation and regression analysis, we estimated the degree of influence of the studied factors on the competitiveness of corn. The results of the analysis are presented in Table 7.

**Table 7.** Results of the regression analysis of competition factors of corn in Russia, Kazakhstan, and Belarus from 2010 to 2016.

| Factor | Coefficient | Standard Error | *t*-Statistic | Prob (F-Statistic) |
|---|---|---|---|---|
| | | Russian Federation | | |
| Constant | −2.09 | 0.135 | −1.55 | 0.037 |
| Average prices (APV) | −0.08 | 0.002 | −3.18 | 0.019 |
| Quantity (Q) | 0.08 | 0.003 | 3.06 | 0.02 |
| Export prices (EX_P) | −0.11 | 0.000 | 3.22 | 0.019 |
| Share of export (EX_Q) | 0.37 | 0.012 | −3.16 | 0.019 |
| Coefficient of competition (Com_C) | −0.39 | 1.85 | −3.22 | 0.019 |
| $R^2 = 91\%$, F = 20.99, Prob (F) = 0.016. | | | | |
| | | Republic of Belarus | | |
| Constant | 0.25 | 0.01 | 2.53 | 0.024 |
| Average prices (APV) | −0.13 | 0.000 | 2.33 | 0.026 |
| Quantity (Q) | 0.05 | 0.000 | 2.47 | 0.025 |
| Export prices (EX_P) | −0.12 | 0.000 | −2.68 | 0.023 |
| Share of export (EX_Q) | 0.05 | 0.0045 | −0.84 | 0.056 |
| Coefficient of competition (Com_C) | −0.33 | 0.010 | −6.18 | 0.01 |
| $R^2 = 90.5\%$, F=12,22, Prob (F) = 0.017. | | | | |
| | | Republic of Kazakhstan | | |
| Constant | 1.77 | 0.018 | 7.06 | 0.009 |
| Average prices (APV) | −0.80 | 0.00 | −2.72 | 0.022 |
| Quantity (Q) | 0.09 | 0.00 | −1.09 | 0.047 |
| Export prices (EX_P) | −0.26 | 0.00 | −3.19 | 0.019 |
| Share of export (EX_Q) | 0.063 | 0.003 | 1.69 | 0.034 |
| Coefficient of competition (Com_C) | −0.34 | 0.017 | 0.23 | 0.0085 |
| $R^2 = 84.7\%$, F (s) = 5.55, Prob (F) = 0.1. | | | | |

The data in Table 7 shows that the price competitiveness ratio had the greatest impact on the level of corn competitiveness in the Russian Federation. The average producer prices of corn in Russia in 2016 was about USD $125/ton. The second and third places in terms of impact on the competitiveness of Russian corn in 2016 were the share of exports in production, at 35%, and export prices at USD $929/ton, respectively. The volume of production of this crop grew from 3.1 million tons in 2010 to 15.3 million tons in 2016, i.e., by almost five times, and not only satisfied the domestic demand for corn for grain, but also increased the volume of supply of corn to foreign markets from 0.3 to 5.32 million tons in the period. The current prices allow for strong competitive positions

for corn grain produced in Russia, both in the domestic and export markets, for agricultural products to be maintained.

In the Republics of Belarus and Kazakhstan, corn for grain is produced in small volumes, which were 740,000 and 762,000 tons in 2016, respectively. In the period from 2010 to 2016, the volumes of the gross harvest of this crop grew steadily. In Belarus for the period under review, the growth was 34.3% and that in Kazakhstan was 65%. Due to the comparatively small volumes of corn production and export in Kazakhstan, its competitiveness was influenced by average producer prices, whose level was the lowest among all EAEU countries, as well as the ratio of domestic prices to corn prices (the price competitiveness factor). The Republic of Belarus was the least competitive level in both domestic and export prices.

Thus, the proposed econometric model enabled the identification of the competitive advantages of Russia, Kazakhstan, and Belarus in the grain market, and the degree of influence of the studied factors on the competitiveness of wheat, barley, and corn in the domestic EAEU market. To assess the competitiveness of products in foreign markets, we compared domestic producer prices with producer prices in the countries recognized as global leaders in exporting these products. Analysis was conducted for Belarus, Kazakhstan, and Russia (Table 8).

**Table 8.** Average producer prices for agricultural products in the EAEU and the leading export countries in the world in 2016, USD $/ton.

| Product | Belarus | Kazakhstan | Russia | Leading Export Countries In The World | | | | |
|---------|---------|------------|--------|-------------|-------------|---------------|---------------|---------------|
| Wheat | 214 | 105 | 132 | Russia 132 | USA 143 | Australia 206 | Canada 174 | Ukraine 129 |
| Barley | 174 | 77 | 115 | Australia 188 | France 138 | Ukraine 118 | Russia 115 | Germany 139 |
| Maize (corn) | 560 | 114 | 124 | USA 132 | France 176 | Hungary 148 | Ukraine 138 | Russia 124 |

Source: Authors' calculations based on data of Food and Agriculture Organization of the United Nations FAO statistical data [26].

Thus, in Kazakhstan and Russia, the producer prices for wheat, barley, and corn were lower than in the countries who were the main exporters of these products in the world, showing a significant competitive advantage in price. In the Republic of Belarus, producer prices for these crops were higher than in the countries leading in terms of global exports, which means that these products are less competitive in Belarus.

## 5. Conclusions and Proposals

In general, the production of grain in Russia and Kazakhstan, both in the domestic market of the EAEU and in the global market, is competitive. The main factors increasing competitiveness in 2014–2016 were the decrease in domestic producer prices and the growth in production volumes. These two aspects will allow the EAEU countries to gain a larger share in the international grain and processed products markets in the future.

The export of Russian grain is forecasted to reach 40 million tons by 2020 [2]. In Kazakhstan, in 2020, grain exports are expected to be 6 million tons. The share of Belarus in world grain exports is insignificant, but by 2020, grain production will increase up to 10 million tons, which will also increase its export opportunities.

To further increase the production and competitiveness levels for grain and grain processing products in EAEU countries, it is necessary to develop and introduce quality seed, implement more intensive use of new low-cost technologies, improve the price relationships in the grain market, provide state support for the grain subcomplex, harmonize domestic requirements for quality grain with those of foreign grains, expand flour exports and products of deep grain processing,

develop internal and external infrastructure, and improve the transport and logistics system of the grain market in the long-term. In conclusion, a single sustainable strategy must be created and implemented for the development of grain exports in the EAEU.

**Author Contributions:** V.M. formulated the hypothesis, analyzed the data, and wrote the manuscript. N.Z., C.F. and M.A. formulated the theses, analyzed the data, and wrote the manuscript.

**Funding:** This research received no external funding.

**Acknowledgments:** The authors want to thank the post graduate student M. Galkin for his help in preparing the economic-mathematical model.

**Conflicts of Interest:** The authors declare no conflict of interest.

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
