# Peer review of "Competitiveness of Agricultural Products in the Eurasian Economic Union"

_agriculture, doi:10.3390/agriculture9030061_

Round 1

Reviewer 1 Report

Dear editor and authors,

Thank you very for giving me the opportunity of reviewing this paper proposal. In this paper, the authors’ main objective is to investigate the competitiveness of grain and ‘recycling’ in the Eurasian Economic Union with an econometric model. Although the purpose of the paper is interesting, there are several points that need to be revised before publication. I have grouped them under four categories: the purpose of the paper, the clarity of explanations, the critical assessment of information and results, and the econometric model.

Purpose of the paper

The purpose of the paper is, in my opinion, unclearly stated and structured. In the introduction, the authors mention the analysis of the competitiveness of grain and its ‘recycling’, which in section 3, based on the information that is delivered, seems to be understood as pasta and flours. Unfortunately, in section 4, no econometric model analyzes the competitiveness of flour. I understand that the authors have done a lot of work to provide all these information. However, the inconsistency of the different lenses used in the different sections of the paper makes the authors’ message a bit obscure and difficult to understand. In addition, it is never clearly stated why the authors decided to add a focus on pasta and flour. Finally, in the last table, the authors compare agricultural prices. I would suggest the authors streamline their paper on grain competitiveness only. In this case, the similarity of compared commodities (e.g., similarity of market, similarity of production processes – agricultural vs agri-industrial) is consistent all over the paper. Not only will it make the paper clearer to the audience, but it will also enable the authors to deepen their analyses and strengthen their conclusions.

Clarity of explanations

Another limitation of this paper is the lack of explanations. My understanding of the information is that it is left to the reader to make connections, justify authors’ choices, and elicit assumptions. These points make the reading of the paper very difficult. Below are suggestions to improve the clarity of the paper. There is style considerations behind them; therefore, I leave to the authors to choose how they want to improve the clarity of the paper.

-          The background of the paper needs to be strengthened. For instance, definitions can be added, which, in my opinion, is the case when authors speak of implementation mechanism (l.31). The role of the EAEU could be quickly assessed and main measures in relation to agricultural production described. This would enable the authors to discuss policy-making more consistently in a discussion section.

-          References need to be added when strong statements are made (e.g., describing the EAEU). Generally speaking, the paper lacks references that embed their work in the existing literature.

-          In the introduction, the authors make two references to method. Please group them to improve the flow.

-          The authors used several sorts of analysis to infer their conclusions. They are mentioned between l.117/120. However, the articulation of these methods are not explicated. What does each method bring? How are they complementary?

-          In the description of the factors (l.136-159), the authors could introduce sub-sections to either separate each factor or to group them under a common theme.

-          In the following lines (l.160-169), the authors provide information about grain production in the EAEU. Again, the lack of connection of this information with the rest of the article makes the reading difficult. Maybe this information in the introduction to provide more background? Or maybe additional links could be inserted to help the reader? (Same comment for l.170-194-204).

-          The explanations about the econometric model provided between l.209-250 are part of the method.

-          Results significance is not easily accessible to the reader and not clearly mentioned in the text either.

Critical assessment

At each step of the paper, the authors make choices that are not necessarily justified nor critically assessed. This information is critical for the understanding of the reader. Readers may not be as experienced and knowledgeable as the authors. Therefore, it is important to provide such information even though it might seem unnecessary.

-          Between l.53 and l.57, many authors are cited, a list that is followed by several references. The references should be clearly connected to the author they refer to. What is the contribution of each author? This information should be provided to the audience. It is not the role of the reader to do this search.

-          This is also the case in the method section l.137, 142, 148.

-          Between l.63 and l.88, the authors seem to review several methods that could be used to conduct an analysis on the competitiveness of grain. However, they do not critically assess each method. Therefore, it is not clear why they choose the method that they then describe.

-          L.98/99, again, it is not clear why the indicators are divided into two groups, please explain.

-          The results of the econometric analyses are provided by commodity (i.e., wheat, barley, and corn). The results for each commodity are described in a sub-section. This structure prevents a critical assessment of the results. What are the similarities and the differences between each type of commodity? In order to improve this, I would suggest that the authors revise their table and provide only the information necessary for the analysis (less columns), and merge the three tables together. This way, the reader can more easily compare the results. It will also enable the authors to conduct a more critical analysis instead of a description of the results.

-          The last analysis conducted by the authors consist of discussing the competitiveness of grain product on the basis of agricultural prices. The utilization of agricultural prices instead of each commodity price is a discrepancy. There are differences in the results that cannot be discussed and assessed since only one price is used in this analysis.

The econometric models

A major feedback concerns the econometric models:-

-          two models are detailed between l.216 and 250. The relationship between these two models needs to be elicited

-          in the first model, as far as I understand, the dependant variable is the weighted average integral indicator and the independent variables are the country-specific weighted average integral indicators. It is not clear why the authors chose these variables. An average value is necessarily correlated to the values characterizing the individuals that constitute the group. Hence, what is the purpose of this model?

Spelling/Format

-          Consider using member countries instead of countries–members.

-          Itemized information is to be revised (e.g., l.66-74)

Author Response

Response to the comments and recommendations of the first reviewer:

Thank you for your critical reading and constructive suggestions. Find corresponding changes made by the authors below:

1.    Methodology for the assessment of competitiveness was consolidated by Russian and foreign scientists research.

2.    The structure of the article was improved: the method developed by the authors was moved to the section 2 «Methodology».

3.    Detailed explanations on the use of the method for assessing the competitiveness of the agricultural products of the EAEU countries are provided by the authors.

4.    In section 3 «Results and discussions» approbation of the authorial methodology is conducted on the grain market sample (excluding flour and noodles) of the EAEU countries.

5.    Explanations and clarifications on the use of econometric model were given.

Reviewer 2 Report

The paper identifies and assesses the main factors affecting the competitiveness of wheat, barley, corn, wheat flour, pasta in member states of the Eurasian Economic Union (EAEU). An integral index, based on indicators of competitiveness in terms of prices and production volumes, is built for each product, and a correlation-regression analysis determines the main drivers of competitiveness. Findings suggest that grain-based products are competitive in Belarus, Kazakhstan, and Russian Federation, but not in Armenia and Kyrgyzstan. Their competitiveness is mostly driven by the decrease in domestic producer prices and growth in production volumes.

The main drawback of the paper is the completely lack of a background that justifies the importance of the study. What is the gap in literature? How do the Authors contribute to the existing knowledge? What is the novelty of the study?

The paper has too many self-citations (about 30%), while a section of literature review is lacking in the paper. How does previous literature deal with drivers of competitiveness? For instance, price competitiveness may be affected by international price volatility (cfr. Ivanic and Martin, 2014; Ott, 2014; Santeramo et al., 2018; Santeramo and Lamonaca, 2019a), and production and export volumes may be influenced by trade policies (cfr. Anderson and Nelgen, 2012; Dal Bianco et al., 2016; Santeramo and Lamonaca, 2019b, c; Santeramo et al., 2019a, b)

The paper needs further improvements:

- The title and the stated aim of the paper (e.g. lines 36-37) are misleading: they refer to the agri-food products in general, but the paper focuses only on the EAEU grain market.

- The abstract is too vague about results, while provides unnecessary details on data sources (e.g. the description of UN Comtrade dabase).

- Data presented in section 3, related to prices, production volumes, exports, etc., should be

summarized in a table showing average or data-point values for each product in each country.

- Why does the model in equation (7) consider only 3 out of 5 member states? What about Armenia and Kazakhstan (despite they are not major producers – cfr. lines 209-211)? The exclusion of two out of five member states does not allow to generalize results for the EAEU, and the identification of main drivers of competitiveness becomes case-specific.

- The paper should be reorganized as to have sections of literature review, methodology, and results and discussion. For instance, consider to move part of sections 2 (from line 81) and 4 (lines 209-250)

in a specific section of methodology, followed by a description of data (from section 3).

- There are a lot of grammatical errors, typos, (e.g. lines 36-37, 121-123, 190) and repetitive sentences (e.g. cfr. lines 77 and 82-83) all along the manuscript. Please, carefully revise the manuscript.

- The reference list should follow the order of appearance of citations in the text (see author’s

guidelines for Agriculture).

Author Response

Response to the comments and recommendations of the second reviewer:

Thank you for your critical reading and constructive suggestions. Find corresponding changes made by the authors below:

1.    The structure of the article was changed and further developed.

2.    The section “Methodology” was further developed.

3.    The method developed by the authors regarding/on the assessment of competitiveness of the agricultural goods of the EAEU countries can be applied to various types of agri-food products. The choice of ‘the grain market’ as the object of the ‘factor analysis of competitiveness’ is declared in section 1 “Introduction”.

4.    The main producers of grain in the EAEU countries are Russia, Belarus and Kazakhstan. Their combined share in global grain production is almost 99 %, that is why ‘the correlation regression analysis of the competitive factors’ is focused particularly on the grain market of these countries.

5.    The reference list is improved and further developed.

Round 2

Reviewer 1 Report

Dear editor and authors,

I would like to thank the auhors for the deep restructuring of their paper. The content flows well and it is easier to follow the progression of the argument.

I have a few additional minor comments:

-        The term countries-members have not been corrected all over the paper. It would be helpful to homogenize the utilization of the term all over the paper.

-        The term agro/agri-food needs to be changed in agricultural given the restructuration of the paper in most cases of its current utilization

-        L.37: it is sufficient to mention context of globalization, whether it is growing or not seems to be controversial these days

-        L.49-53 the authors construct their argument around grain/cereals but illustrate it with wheat numbers, please provide numbers about grin/cereals

-        Consider using enable instead of allow. Additionally, it is not necessary to add a pronoun after these verbs.

-        L.245-312, this descriptive information would nicely fit in the introduction or be broken down after the discussions related to the five indicators in the methodology section. If it is meant to remain in the results section, then sub-sections can be introduced in the results section.

Author Response

Thank you for your critical reading and constructive suggestions.
Find corresponding changes made by the authors below:

1. The term “EAEU Member States” is changed by the term “EAEU countries”.

2. The term “agri-food products” is changed by the term “agricultural products”.

3. Added data on the proportion of barley of all EAEU countries in the global production and trade.

           4. The section ‘Results and discussions’ are divided into subsections.

Reviewer 2 Report

None

Author Response

Thank you!